# PARP1 Characterization as a Potential Biomarker for *BCR::ABL1* p190+ Acute Lymphoblastic Leukemia

**DOI:** 10.3390/cancers15235510

**Published:** 2023-11-22

**Authors:** Caio Bezerra Machado, Emerson Lucena da Silva, Wallax Augusto Silva Ferreira, Flávia Melo Cunha de Pinho Pessoa, Andreza Urba de Quadros, Daianne Maciely Carvalho Fantacini, Izadora Peter Furtado, Rafaela Rossetti, Roberta Maraninchi Silveira, Sarah Caroline Gomes de Lima, Fernando Augusto Rodrigues Mello Júnior, Aline Damasceno Seabra, Edith Cibelle de Oliveira Moreira, Manoel Odorico de Moraes Filho, Maria Elisabete Amaral de Moraes, Raquel Carvalho Montenegro, Rodrigo Monteiro Ribeiro, André Salim Khayat, Rommel Mário Rodriguez Burbano, Edivaldo Herculano Correa de Oliveira, Dimas Tadeu Covas, Lucas Eduardo Botelho de Souza, Caroline de Fátima Aquino Moreira-Nunes

**Affiliations:** 1Department of Medicine, Pharmacogenetics Laboratory, Drug Research and Development Center (NPDM), Federal University of Ceará, Fortaleza 60430-275, CE, Brazil; caio.bmachado97@alu.ufc.br (C.B.M.); lucenaemerson@alu.ufc.br (E.L.d.S.); flaviamelocpp@alu.ufc.br (F.M.C.d.P.P.); odorico@ufc.br (M.O.d.M.F.); betemora@ufc.br (M.E.A.d.M.); rmontenegro@ufc.br (R.C.M.); 2Laboratory of Cytogenomics and Environmental Mutagenesis, Environment Section (SAMAM), Evandro Chagas Institute (IEC), Ananindeua 67030-000, PA, Brazil; wallaxaugusto@gmail.com (W.A.S.F.); ehco@ufpa.br (E.H.C.d.O.); 3Center for Cell-Based Therapy, Regional Blood Center of Ribeirão Preto, University of São Paulo, Ribeirão Preto 14051-140, SP, Brazildaianne.carvalho@hemocentro.fmrp.usp.br (D.M.C.F.); izadorapf@usp.br (I.P.F.); rafaelarossetti@usp.br (R.R.); msilveira.roberta@usp.br (R.M.S.); sarah.caroline.lima@usp.br (S.C.G.d.L.); dimas@fmrp.usp.br (D.T.C.); lucas.souza@hemocentro.fmrp.usp.br (L.E.B.d.S.); 4Molecular Biology Laboratory, Ophir Loyola Hospital, Belém 66063-240, PA, Brazil; fernando.mellojr@hotmail.com (F.A.R.M.J.); line.seabra@gmail.com (A.D.S.); rommel@ufpa.br (R.M.R.B.); 5Institute of Health and Biological Studies, Federal University of the South and Southeast of Pará, Marabá 68501-970, PA, Brazil; cibelle@unifesspa.edu.br; 6Department of Hematology, Fortaleza General Hospital (HGF), Fortaleza 60150-160, CE, Brazil; rmonteiroribeiro@gmail.com; 7Department of Biological Sciences, Oncology Research Center, Federal University of Pará, Belém 66073-005, PA, Brazil; khayatas@gmail.com; 8Northeast Biotechnology Network (RENORBIO), Itaperi Campus, Ceará State University, Fortaleza 60740-903, CE, Brazil

**Keywords:** poly(ADP-ribose) polymerase inhibitors, acute lymphoblastic leukemia, molecular targeted therapy, drug repositioning

## Abstract

**Simple Summary:**

Acute lymphoblastic leukemia (ALL) is the most common childhood cancer and the presence of *BCR::ABL1* fusion in p190 isoform is a marker for worse prognosis, associated with treatment resistance and reduced overall survival. We aim to determine the effectiveness of targeting poly-ADP-ribose polymerase (PARP) in a model of *BCR::ABL1* p190+ ALL using a small molecule inhibitor, AZD2461, and we hope that our findings may help improve molecular stratification and prognosis of ALL patients.

**Abstract:**

Detection of t(9;22), and consequent *BCR::ABL1* fusion, is still a marker of worse prognosis for acute lymphoblastic leukemia (ALL), with resistance to tyrosine-kinase inhibitor therapy being a major obstacle in the clinical practice for this subset of patients. In this study, we investigated the effectiveness of targeting poly-ADP-ribose polymerase (PARP) in a model of *BCR::ABL1* p190+ ALL, the most common isoform to afflict ALL patients, and demonstrated the use of experimental PARP inhibitor (PARPi), AZD2461, as a therapeutic option with cytotoxic capabilities similar to that of imatinib, the current gold standard in medical care. We characterized cytostatic profiles, induced cell death, and biomarker expression modulation utilizing cell models, also providing a comprehensive genome-wide analysis through an aCGH of the model used, and further validated PARP1 differential expression in samples of ALL p190+ patients from local healthcare institutions, as well as in larger cohorts of online and readily available datasets. Overall, we demonstrate the effectiveness of PARPi in the treatment of *BCR::ABL1* p190+ ALL cell models and that PARP1 is differentially expressed in patient samples. We hope our findings help expand the characterization of molecular profiles in ALL settings and guide future investigations into novel biomarker detection and pharmacological choices in clinical practice.

## 1. Introduction

Acute Lymphoblastic Leukemia (ALL) is a hematopoietic malignancy characterized by the abnormal clonal expansion of lymphocytic lineage cells, leading to an increase in the frequency of blasts in the bone marrow and the accumulation of B or T lymphocytes in the peripheral blood, or more rarely, of natural killer lymphocytes [1]. The worst prognosis for ALL patients is seen in the presence of t(9;22)(q34;q11), which originates the Philadelphia chromosome (Ph), and at the breakpoint, the chimeric gene *BCR activator of RhoGEF* and *GTPase::ABL proto-oncogene 1* (*BCR::ABL1*) [2].

Depending on the breakpoint in the *BCR* gene, the resulting *BCR::ABL1* oncoprotein will have three possible isoforms, one is the classic translocation of 210 kilodaltons (p210), and the others are the non-classic isoforms of 190 (p190) or 230 (p230) kilodaltons [3]. While *BCR::ABL1*-positive tumors had their treatment revolutionized by the development of tyrosine-kinase inhibitors (TKI), many patients with Ph+ tumors still struggle with the emergent cases of therapeutic resistance, especially when afflicted with the p190 isoform, which is much less responsive to TKI treatment [4,5].

Regardless of the isoform, however, the presence of *BCR::ABL1* in malignant cells is highly associated with increased genomic instability due to the cytoplasmic accumulation of reactive oxygen species through the deregulation of the mitochondria membrane potential and with the promotion of non-conservative DNA repair pathways, such as non-homologous end-joining, being responsible for an increase in tumor mutational burden and accelerated leukemic progression [6,7,8].

In the oncological practice, the over reliance of tumors on non-conservative repair pathways has been explored as a therapeutic strategy through the pharmacological inhibition of poly-ADP-ribose polymerase-1 (PARP1), representing the first clinical use of the concept of synthetic lethality [9]. PARP1 is the main effector of the PARP family and is responsible for signaling DNA damage and recruiting enzymes of the DNA damage repair (DDR) machinery, acting through post-translational auto-modifications called PARylation [10,11].

Recently, clinical studies investigated the use of PARP inhibitors (PARPi) which are less susceptible to multidrug resistance such as AZD2461, a structural analogue of olaparib with pan-PARP activity and with small changes in functional groups that allow for the decreased affinity of interaction with transmembrane efflux pumps [12].

Although well established in the treatment of gynecological tumors, clinical investigations regarding the efficacy of PARPis for the treatment of hematological tumors are still sparse, with published results only for the inhibitors olaparib and veliparib. The reported clinical outcomes, however, are encouraging and point towards modest improvements in patient prognosis, with the main advantage of having low toxicity profiles that allow for the use of combination treatment strategies [13,14].

In this study, we measured *PARP1* expression levels in cell models representative of acute lymphoblastic malignancies and aimed to investigate the efficacy of AZD2461 in inhibiting cell growth, inducing cell death as well as modulating biomarker expression in a model of ALL with *BCR::ABL1* p190+ isoform, comparing the cytotoxicity profiles of AZD2461 with imatinib, the *BCR::ABL1* inhibitor which is a gold-standard in the clinical practice for the treatment of Ph+ tumors. We also characterized the cell line SUP-B15, our model of ALL p190+, regarding chromosomal functional gains and losses through an aCGH array and aimed to establish a correlation between the presence of the *BCR::ABL1* p190+ isoform and *PARP1* expression levels in patient samples with ALL.

## 2. Materials and Methods

### 2.1. Cell Culture

The cell line model for ALL p190+ of B-lymphocytes, SUP-B15, was maintained in RPMI medium supplemented with 20% fetal bovine serum (FBS), while the cell lines representative of models of Burkitt’s lymphoma, with a phenotype of B-lymphoblasts, Raji and Namalwa, that were used as comparative controls, were maintained in RPMI medium supplemented with 10% FBS. The non-neoplastic cell line MRC5 used as comparative control for the expression analysis was maintained in DMEM medium supplemented with 10% FBS. All cell lines were acquired directly from the American Type Culture Collection (ATCC^®^, Manassas, VA, USA).

### 2.2. Chemical Treatments

The drugs Imatinib Mesylate (Ca. SML1027) and Doxorubicin Chlorohydrate (Ca. D1515), as well as the molecule AZD2461 (Ca. SML1858), were obtained through the company Sigma-Aldrich (St. Louis, MO, USA) and diluted in Dimethyl Sulfoxide (DMSO) for use.

### 2.3. Alamar Blue Cytotoxicity Assay

Cells were plated in 96-well plates in concentrations ranging from 1.5 × 10^4^ to 2 × 10^4^ cells/well in 100 µL of medium and maintained in culture for 24 h at 37 °C, after which the wells were then treated with another 100 µL of medium containing either imatinib, AZD2461, or doxorubicin in serial dilutions of 10 μM, 2 μM, 0.4 μM, 0.08 μM, 0.016 μM, and 0.0032 μM. After treatment, cells were incubated for 24, 48, and 72 h, and Alamar Blue solution (0.02%) was added after each time point, with further incubation ranging from 3 to 5 h. The fluorescence was measured by spectrophotometry with an excitatory wave length of 560 nm and an emission wave length of 590 nm [15]. A blank well with no cells and a well with no addition of drugs were added for each experiment, representing the negative and positive growth controls, respectively.

To minimize drug-cell exposure time, the subsequent experiments were performed based on the 24-h minimum inhibitory concentration for 50% of the cells (IC50). Sub-inhibitory concentrations were used, aiming to determine how the proposed treatments alter cell metabolism prior to apoptosis induction.

#### Alamar Blue Statistical Analysis

Each experiment was performed in triplicates and the data were analyzed through non-linear regression of the inhibition percentual vs the concentration log to determine the IC50 and the respective confidence intervals (IC 95%) through the software GraphPad Prism (version 5.01).

### 2.4. Flow Cytometry Analysis

For all assays analyzed through flow cytometry, cells were plated in 12-well plates in a density of 2 × 10^5^ cells/well in 1 mL of RPMI medium and maintained in culture for 24 h at 37 °C, after which the wells were then treated with 500 µL of medium containing either imatinib or AZD2461 in 24-h sub-inhibitory concentrations of 1.5 μM. The plates were then re-incubated in the same parameters for another 24 h. A non-treated control was used in all experiments, with the addition of 500 µL of medium containing an equivalent volume of DMSO. After the 24-h treatment, cells were counted in hemocytometer to ensure that no significant cell death was detected, being then pelleted and the supernatant was removed.

#### 2.4.1. Cell Cycle Arrest

Cells were fixed and permeabilized utilizing the kit eBioscience™ Foxp3/Transcription Factor Staining Buffer Set (ThermoFisher Scientific^®^, Waltham, MA, USA), following the instructions provided by the manufacturing company. Cells were resuspended in a RNAse solution containing phosphate-buffered saline (PBS) 1×, 5% FBS, and propidium iodide to a final concentration of 25 µg/mL. Resuspended cells were incubated for 2 h in the dark and then pelleted and resuspended in 400 µL of PBS 1× solution and the fluorescence intensity was analyzed in the flow cytometer BD FACSymphony^TM^ (BD Biosciences, Franklin Lakes, NJ, USA).

#### 2.4.2. Induction and Quantification of Early Apoptosis

A work solution of annexin-V binding buffer was prepared from the 10× Annexin-V Binding Buffer (Invitrogen™, Waltham, MA, USA) following the manufacturer-provided protocols and cells were resuspended in 100 µL of this solution, after which 5 µL of the reagent Annexin V FITC conjugate (Invitrogen™) was added, and cells were incubated in the dark for 15 min at room temperature. After the incubation, 5 µL of 7-aminoactinomycin D (7-AAD) was added as a counterstain to differentiate early and late apoptotic cells and the samples were resuspended to a final volume of 400 µL of annexin-V binding buffer work solution and the fluorescence intensity was analyzed in the flow cytometer BD FACSymphony^TM^ (BD Biosciences, Franklin Lakes, NJ, USA).

#### 2.4.3. Alterations in PARP1 Intracellular Levels

Cells were fixed and permeabilized utilizing the kit eBioscience™ Foxp3/Transcription Factor Staining Buffer Set (ThermoFisher Scientific^®^), following the instructions provided by the manufacturing company. The Recombinant Alexa Fluor^®^ 647 Anti-PARP1 antibody [E102], acquired through Abcam^®^ (Boston, MA, USA), was diluted in PBS 1× solution in a 1:1000 ratio, as well as the isotype control Recombinant Alexa Fluor^®^ 647 Rabbit IgG monoclonal [EPR25A], being added to the samples and incubated for 30 min in the dark at room temperature. Samples were pelleted and resuspended in 400 µL of PBS 1× solution and the fluorescence intensity was analyzed in the flow cytometer BD FACSymphony^TM^ (BD Biosciences, Franklin Lakes, NJ, USA).

#### 2.4.4. Flow Cytometry Statistical Analysis

The emitted fluorescence was analyzed through ten thousand events detected through flow cytometry (BD FACSymphony^TM^, BD Biosciences, Franklin Lakes, NJ, USA). Data were examined from the mean and standard deviation of three individual experiments utilizing either the Modfit LT^TM^ (version 5.0) or the FlowJo (Version 10.8.1) software. To detect significant differences between the control and treated groups, the data were compared through analysis of variance (ANOVA) followed by Bonferroni’s post-test. Significant differences were considered with an interval of confidence of 95% (*p* < 0.05). GraphPad Prism 5.01 software (Merck^®^, Darmstadt, Hesse, Germany) was used for data analysis and graph design.

### 2.5. Expression Analysis

Total RNA was extracted with TRIzol Reagent^®^ (Invitrogen™, Carlsbad, CA, USA), following the instructions provided by the manufacturing company. Extracted RNA was quantified utilizing the equipment NanoDrop (Thermo Scientific, Carlsbad, CA, USA) and 20 ng was used for cDNA confection using the High-Capacity cDNA Reverse Transcriptase kit (Life Technologies, Carlsbad, CA, USA).

Primers for the *PARP1* (Reference: Hs00242302_m1), *BCR::ABL1* p190 (e1a2) (Reference: Hs03024844_ft), and *ACTB* (Reference: Hs01060665_g1) genes were synthesized by the company ThermoFisher^TM^ Scientific and real-time quantitative PCR (qPCR) reactions were carried out through the TaqMan^®^ Gene expression assays (Applied Biosystems^®^, Waltham, MA, USA). Amplified fragments were quantified in the equipment QuantStudio^®^ 5 (Applied Biosystems^®^).

#### Gene Expression Statistical Analysis

Experiments were performed in triplicates and the relative expression levels were calculated through the 2^−ΔΔCT^ method, utilizing either the non-neoplastic cell line MRC5 or the non-treated controls as calibrators [16,17]. Data were compared through one-way ANOVA followed by Bonferroni’s post-test. Significant differences were considered with an interval of confidence of 95% (*p* < 0.05). GraphPad Prism 5.01 software (Merck^®^, Darmstadt, Hesse, Germany) was used for data analysis and graph design.

### 2.6. Chromosomal Analysis and Gene Ontology (GO)

#### 2.6.1. DNA Extraction

Cell line DNA was extracted using the Wizard^®^ Genomic DNA Purification kit (Promega Corporation, Madison, WI, USA), and purity and integrity were assessed on the Agilent 2200 TapeStation (Agilent Technologies, Santa Clara, CA, USA) with D1000 ScreenTape (Agilent Technologies), according to the respective manufacturer’s protocol. Only samples with DNA Integrity Number (DIN) > 7 were used for downstream analyses.

#### 2.6.2. Array-Based Comparative Genomic Hybridization (aCGH) Analysis

Array-CGH (aCGH) experiments were performed on an Agilent microarray platform (Agilent Technologies) with a SurePrint G3 Cancer CGH + SNP Microarray 4 × 180 K slide (Agilent). Sample preparation, labelling, and microarray hybridization were performed according to the Agilent CGH Enzymatic Protocol version 7.5. Slides were scanned using the Agilent G2565CA scanner. Data were extracted with Feature Extraction software (v9.1 Agilent Technologies) and analyzed with Genomic Work Bench 11.0.1, Agilent Cytogenomics 5.0 and GeneSpring GX 14.5, as described elsewhere [18]. The algorithm used was Aberration Detection Method 2 (ADM-2), applying the following filters: threshold = 6; minimum number of probes in region = 3; and Log2Ratio > 0.25 and log2Ratio < −0.25 were defined as copy number gains and losses, respectively. The ideogram showing the identified copy number aberrations (CNAs) was constructed using the PhenoGram online software (https://visualization.ritchielab.org/phenograms/plot (accessed on 6 February 2023)) [19].

#### 2.6.3. Biological Interpretation and Statistical Analysis

Functional enrichment analysis of the CNAs was performed using the R package g:Profiler2 [20] for gene ontology (BP: biological processes; CC: cellular components; MF: molecular function). We used the g:SCS (Set Counts and Sizes) algorithm for the multiple testing correction. *p*  <  0.05 was considered as statistically significant. We used a Manhattan plot to visualize the enrichment results.

### 2.7. Patient Samples

Blood samples were collected from adult patients either at the time of their diagnosis, with ALL or CML, or during medical treatment at the Hospital Geral de Fortaleza (HGF) and at the Ophir Loyola Hospital. This study was approved by the Ethics Committee of the Federal University of Ceará (approval number: 5.207.521) and of the Ophir Loyola Hospital (approval number: 4.409.317), informed written consent was obtained from the patient being brought into the study only after understanding and accepting the terms, and all methods were carried out in accordance with Helsinki guidelines and regulations. A total of 60 patient samples were deemed positive for the presence of *BCR::ABL1* p190 through analysis of the buffy coat genetic material and had their *PARP1* expression levels also quantified. All molecular assays went as previously described in Topic 2.5 utilizing RNA from the buffy coat of either peripheral blood or bone marrow. Immunophenotypic and karyotype data were acquired directly from each patient’s medical records.

#### Statistical Analysis of Patient Samples

Gene expression was quantified as described in Topic 2.5.1. To determine the significance of *PARP1* expression, analyses were made by dividing the cohorts into either ALL or CML patients, and data were analyzed through an average of the fold change expression of the patients in each cohort. Comparisons were made with blood samples from 10 healthy controls utilizing Student’s *t*-test for statistical analysis.

### 2.8. Data Analysis Based on the Gene Expression Omnibus (GEO) Database

We used the Gene Expression Omnibus (GEO) (https://www.ncbi.nlm.nih.gov/gds (accessed on 6 February 2023)), a public repository of high-throughput gene expression data, to profile *PARP1* expression in ALL. We downloaded and analyzed, as described elsewhere [21], the GSE13159 dataset (MILE study) [22,23], which employed Affymetrix HG-U133 Plus 2.0 GeneChips and included 2096 samples.

#### Statistical Analysis for GEO Expression

Statistical analysis was performed using the R language (v.4.3.2). T-tests or Wilcoxon tests were performed for analyses with two groups, and the significance level was set at *p* < 0.05.

### 2.9. Single-Cell Sequencing Analysis

To extend our analysis of *PARP1* expression to the single-cell level, we used the Tumor Immune Single Cell Hub 2 (TISCH2) database [24]. We selected three single-cell RNA-seq (scRNA-seq) datasets: (i) the GSE132509 dataset [25], which consisted of 37,936 cells from 11 childhood acute lymphoblastic leukemia (cALL) patients (eight cALL patients with >50% blasts and three healthy donors); (ii) the GSE153697 dataset [26], which included 2904 cells from one child diagnosed with B-ALL and treated with anti-CD19 CAR-T therapy; and (iii) the GSE154109 dataset [27] comprising 10,800 cells from seven pediatric patients with B-ALL. We downloaded and analyzed these datasets according to the methods previously described [24,28,29].

## 3. Results

### 3.1. Significantly Different PARP1 Expression Levels Are Detected among Lymphoblastic Cell Lines

Comparative expression of *PARP1* in leukemic cell lines relative to the non-neoplastic cell model reveals significant overexpression in both SUP-B15 (*p* < 0.0001) and Raji (*p* < 0.05), with incremental fifteen- and seven-fold changes in expression, respectively, while the *PARP1* expression level in the cell line Namalwa was very similar and not statistically different from the control (Figure 1). SUP-B15 significantly overexpressed *PARP1* when compared with either of the neoplastic cell lines (*p* < 0.01), with Raji also presenting significantly higher expression when compared with Namalwa (*p* < 0.05).

### 3.2. PARP Inhibitor Is highly Cytotoxic to the BCR::ABL1 p190+ Cell Line

Table 1 describes the IC50 of AZD2461, imatinib, and doxorubicin after 72 h of incubation with the cell lines SUP-B15, Raji, and Namalwa. AZD2461 is the structural analogue of Olaparib with pan-PARP inhibitory activity and with modifications in functional groups that lessen its affinity of interaction with transmembrane efflux pumps of the ATP binding cassette subfamily B (ABCB) family [12]. Imatinib is the TKI with inhibitory activity over BCR::ABL1 chimeric protein and is considered the gold standard for the treatment of Ph+ tumors [30]. Finally, doxorubicin is a well-established and highly cytotoxic chemotherapeutic drug with action over rapidly dividing cells due to the ability to inhibit topoisomerase activity [31], and it was used as a comparative control to the inhibition values of the targeted therapies.

The use of doxorubicin, as expected, was met with significantly lower values of IC50 than the molecular targeted therapies due to the non-specificity of its action over the viability of rapidly dividing cells [32], presenting cytotoxic activity in the order of 10^1^ nanomolar (nM).

In the SUP-B15 cell line, the IC50 values of AZD2461 and imatinib were low and extremely similar to one another, demonstrating almost equivalent inhibition potential. Due to harboring *BCR::ABL1* translocation, the inhibitory activity of imatinib over SUP-B15 is not a surprise and has already been described in the literature [33]; however, the inhibitory concentration achieved with AZD2461 of 344.3 nM in 72 h, close to the 329.2 nM of the inhibition of imatinib, suggests that the use of PARPis may be as effective as the treatment with TKI in models of ALL p190+.

In the cell lines Raji and Namalwa, used as comparative controls of other B-cell neoplasms with different PARP1 expression levels, the 72-h IC50 of AZD2461 and imatinib showed great disparity. In both cell lines, imatinib presented some cytotoxic activity, probably due to its non-specific activity over other tyrosine kinase pathways that are essential in the maintenance of proliferative capabilities and in the survival of neoplastic clones [34]. However, the use of AZD2461 did not present relevant inhibitory capabilities, with an IC50 of over 20 µM for Namalwa while not even being able to reach an IC50 for Raji in the concentrations used in this assay.

Due to the non-responsiveness of Raji and Namalwa to the proposed treatment, cytotoxic assays of 48 and 24 h of treatment were only carried out with the SUP-B15 cell line, as well as all the subsequent phenotypic and molecular characterization assays. The obtained results are described in Table 2 and the inhibition patterns mimic those observed in the 72-h treatments, with doxorubicin presenting considerably higher cytotoxic activity in comparison to the targeted therapies and an equivalence in the inhibitory potential of AZD2461 and imatinib.

Taking into consideration the three treatment intervals, the inhibitory values of AZD2461 and imatinib in incubation with SUP-B15 were not significantly different when analyzed through Student’s *t*-test, demonstrating the equivalence of both drugs in treating this ALL p190+ model in vitro.

### 3.3. AZD2461 Promotes Similar Cell Cycle Arrest Profile to That of Imatinib in SUP-B15

Analysis of cell cycle arrest induced by treatment strategies allows for the identification of a drug’s potential cytostatic properties, inhibiting neoplastic clones proliferation [35,36,37]. Figure 2 describes cell cycle arrest profiles of either AZD2461 or imatinib in 24-h sub-inhibitory treatment of the SUP-B15 cell line in comparison to the non-treated control, and very similar patterns of inhibition may be seen, with considerable cell cycle arrest at the G0/G1 phase (*p* < 0.0001).

In our non-treated control experiment, most events of the SUP-B15 cell line were detected on the G0/G1 phase—47.37%—with a great amount also detected on S phase—43.98%—and only a minority of cells undergoing mitosis on phase G2/M—8.66%. This low presence of SUP-B15 cells in the mitotic phase goes along with what is already described in the literature and is representative of the extensive duplication time of SUP-B15 of approximately 46 h [38]. Furthermore, a tumor sub-population of apparently hypodiploid cells, represented by the yellow peaks in Figure 2, was detected in the analysis of the control cell cycle, constituting 4.21% of events, although the percentage of this population in each phase of the cell cycle was not able to be determined due to fluorescence overlap with the pseudodiploid population of greater frequency.

In the experiments of AZD2461 or imatinib treatment, a significant arrest in phase G0/G1 was observed in comparison to the non-treated control (*p* < 0.0001), with 72,6% and 75,12% of events in G0/G1 to the treatments, respectively, and no statistically significant difference was detected in the arrest induced between both treatments. In both treatments, the arrest was induced with the detriment of S phase cells, being significantly decreased in comparison to the control (*p* < 0.0001), with 19.3% of events in the S phase for the AZD2461 treated experiment and 14.9% of events in S phase for the imatinib treated experiment, once again there not being detected a statistically significant difference between S phase events of both treatments. Finally, in the analysis of the G2/M phase, no difference was detected comparing either treatment with the control, with 8.36% and 9.98% of cells in the G2/M phase for AZD2461 and imatinib, respectively.

Moreover, the same tumor sub-population of hypodiploid clones detected in the control experiment was also observed with a greatly defined fluorescence peak in the imatinib-treated experiment, representing 8.92% of total events, while almost not detected after AZD2461 treatment, representing only 1.7% of events. When submitted to analysis of variance, however, the differences in sub-population frequency did not show statistical difference.

The presence of tumor subpopulations in cell line models is well characterized in the literature due to the high frequency of mutations in neoplastic clones [39]. Although it was not possible to differentiate and molecularly characterize both populations identified in this study, the hypodiploid subpopulation seems highly sensible to treatment with AZD2461, as seen in the diminished population frequency and increase in cell debris, while after imatinib treatment the subpopulation shares the global arrest profile, with an increase of cells in G0/G1, indicating that these hypodiploid cells are more sensible to the cytotoxicity of AZD2461.

The presented data indicates that PARP1 inhibition through the use of AZD2461 has cytostatic capabilities comparable to those of imatinib treatment in ALL p190+ cell models, inhibiting neoplastic clone proliferation due to G0/G1 phase arrest. A tendency for AZD2461 to be cytotoxic to still uncharacterized tumor sub-populations, even in sub-inhibitory concentrations, was also observed, although further analyses are needed to elaborate on this hypothesis.

### 3.4. AZD2461 Induces Early Apoptotic Markings in the SUP-B15 Cell Line after Sub-Inhibitory Treatment

Fluorescent marking with annexin-V reagent conjugated with fluorescein isothiocyanate (FITC) fluorochrome was used to identify exposed phosphatidylserine, a marker of early apoptosis, on the outer membrane of cells after the treatment incubation period [40,41]. Additionally, 7-AAD, a DNA intercalating agent that is unable to permeate the cytoplasm of cells with intact cytoplasmic membrane, was used as a counterstain, and events marked with both fluorochromes represent late apoptotic/necrotic cells [42,43]. Events not marked with any fluorochromes represent viable non-apoptotic cells (Figure 3).

The use of imatinib in sub-inhibitory concentrations of 1.5 µM did not have any effect of apoptosis induction after 24 h of incubation, not being statistically different from the non-treated control experiment. Meanwhile, the use of AZD2461 in the same concentration was able to induce a considerable increase in the frequency of both early apoptosis and late apoptosis/necrosis populations in comparison to control and to cells treated with imatinib (*p* < 0.0001).

The results indicate that AZD2461 acts over apoptosis induction much more prematurely than imatinib in this model of ALL p190+; however, data interpretation must take into account the aforementioned presence of a tumor hypodiploid subpopulation and the potential cytotoxic effect of AZD2461 over this population, raising the hypothesis if AZD2461 actually induces early apoptosis over the cell line as a whole or over specific and differentiated neoplastic clones.

Nevertheless, AZD2461 potential to induce higher rates of apoptosis than the treatment nowadays considered to be the gold standard, imatinib, represents an advance into alternative and complementary therapies for a subset of hematological malignancies that struggle with recurrent cases of tumor resistance and refractoriness.

### 3.5. BCR::ABL1 p190 Expression Is Modulated by Treatment with AZD2461 or Imatinib

Aiming to determine if the proposed treatments influence biomarker expression levels, a qPCR assay of the treated samples quantifying mRNA for *BCR::ABL1* p190+, a hallmark of the SUP-B15 cell line, and *PARP1* was performed, concomitant with PARP1 protein fluorescent marking through flow cytometry.

qPCR analysis reveals that both treatments with AZD2461 (*p* < 0.01) or imatinib (*p* < 0.0001) significantly increase *BCR::ABL1* p190 expression levels in comparison to the non-treated SUP-B15 control group, 3- and 4.5-fold, respectively (Figure 4). Expression upregulation between both treatments was also significantly different (*p* < 0.05), although the same tendency for *BCR::ABL1* p190 mRNA increase was detected. While *BCR::ABL1* overexpression predicts TKI-resistant phenotypes [44], imatinib-induced *BCR::ABL1* upregulation has been previously reported in CML cell models and is associated with neoplastic attempts to evade cell death [45], with our findings expanding this concept of *BCR::ABL1* upregulation as a compensatory mechanism to Ph+ ALL as well. The finding that AZD2461 also upregulates *BCR::ABL1* p190 expression levels reveals a similar pattern of cell death evasion for both treatments.

Regarding *PARP1* expression, however, neither treatment had any statistically significant modulatory effects, remaining close to control mRNA transcript levels. The catalytic activity of PARP1 has already been proposed as the own regulator of *PARP1* mRNA expression [46], and while our data does not indicate the use of PARPi in sub-inhibitory concentrations to have any effect on *PARP1* mRNA expression, high-dose olaparib treatment does downregulate *PARP1* both in mRNA and protein levels [47].

To prove if AZD2461 or imatinib treatments induced post-transcriptional modifications on PARP1 levels, cells were stained with anti-PARP1 antibodies and analyzed by flow cytometry for mean fluorescence intensity (MFI) quantification (Figure 5). Comparison of relative MFIs reveals increased PARP1 in cell populations treated with AZD2461 (*p* < 0.05), while imatinib-treated populations had no statistical difference to the control group. A tendency for lower levels of PARP1 may be seen in the imatinib-treated group, however, as evidenced by an increase in statistical significance when comparing both treated groups to each other (*p* < 0.01).

A clear trail may be seen in the AZD2461-treated graph, revealing a small percentage of events with depletion of PARP1 levels, and taken together results from the expression and PARP1 intracellular levels which seem contrasting. However, our flow cytometry analysis indicates that AZD2461 acts directly over PARP1 protein, inducing degradation and consequent loss of catalytic function, much sooner than it may act over gene expression.

### 3.6. Array Comparative Genomic Hybridization (aCGH) Revealed That PARP1 Is Amplified in the SUP-B15 Cell Line

In order to further characterize the cell model prioritized in this study, we conducted a comprehensive genome-wide analysis using aCGH to screen copy number alterations (CNAs) in the SUP-B15 cell line. Interestingly, this cell line harbored 28 CNAs (4 gains and 24 losses) with sizes ranging from 0.11 Mb to 86 Mb (Figure 6; Appendix A). Large CNAs (>20 kb) were preferentially observed on chromosomes 1, 4, 8, and 14. In addition to these broad CNAs, we also identified numerous localized gains and losses. Some of these affected chromosomal regions where genes were previously shown to be altered in B-ALL, such as *PAX5* at 9p13.2 (loss) [48,49,50], *CDKN2A/B* at 9p21.3 (loss) [49,50], and *MYC* at 8q24.21 [51] (Appendix A). Interestingly, the SUP-B15 cell line harbors a gain in the *PARP1* gene, localized in chromosome 1q42.12, supporting its high mRNA expression observed in the real-time PCR data (Figure 1).

Further, these CNAs were subjected to functional annotation by the g:Profiler2 [20] tool to predict the biological processes in which they were involved. Gains were associated with immune system activation, encompassing immunoglobulin binding and complex formation, antigen binding, complement activation and mediation of immune response, and transmembrane signaling (Figure 7A). In contrast, losses are primarily associated with hemoglobin metabolism through oxygen-carrying capacity, haptoglobin binding, and hemoglobin complex formation (Figure 7B). Overall, our findings contribute to a deeper understanding of the functional consequences of CNAs in B-ALL, revealing their involvement in immune system regulation and hemoglobin metabolism. This knowledge can serve as a foundation for further investigations, ultimately advancing our understanding of the molecular mechanisms underlying these biological processes and potentially providing insights into developing targeted interventions and therapies.

### 3.7. Differential PARP1 Expression Profile among p190+ ALL and CML Patients and Dataset Analysis

Attempting to gather evidence of PARP1 relevance in the landscape of *BCR::ABL1* p190+ tumors in the clinical practice, a cohort of 60 patient samples, including both ALL and CML patients, all of which were quantified and attested as positive for the *BCR::ABL1* p190 isoform, had their *PARP1* expression measured through qPCR. Overall patient’s characteristics, including available age, white blood cell (WBC) count, immunophenotype, karyotype, and qPCR cycle threshold (Ct), are reported in the Appendix A.

Analyzing ALL p190+ patient samples, clear differential expression of *PARP1* may be seen, as evidenced by the significant twofold increase compared with healthy donors (*p* < 0.01; Figure 8A), while statistical analysis of CML p190+ patient samples highlights repressed expression, evidenced by a 0.35-fold change when compared with healthy donors (*p* < 0.01; Figure 8B). To corroborate with our findings from the patient sample analysis, data from the Microarray Innovations in Leukemia (MILE) study were gathered and stratified to discriminate between *PARP1* levels in B-ALL with t(9;22), T-cell ALL, and samples from the bone marrow of healthy donors (Figure 8C). Indeed, the findings support our claim that *PARP1* is overexpressed in Ph+ ALL and also show that this is not the case for T-ALL by tying it to the occurrence of B-cell phenotypes.

Furthermore, investigations into single-cell transcriptomics utilizing available online datasets [25,26,27] further stratify the overexpression of *PARP1* in cohorts of c-ALL cases (Figure 9). The generated heatmap brings into evidence the naturally occurring higher *PARP1* expression in B-cells when compared with other non-neoplastic hematopoietic cell subtypes, being only surpassed by the expression in erythroid progenitor populations of the GSE154109 dataset (Figure 9A), which may be respectively explained by the relevant role of the PARP family in regulating B-cell differentiation, but also being critical for regulating mitosis and rapidly dividing cells as a whole [52,53]. Moreover, the single-cell visualization of subpopulations highlights the overexpression of *PARP1* in the analyzed datasets as properly coming from the malignant clones, instead of other fractions of hematopoietic cells (Figure 9B).

The results highlight that *PARP1* increased expression may not be solely associated with *BCR::ABL1* p190 presence, but rather with the pathogenesis of B-ALL in general. This data also suggests that *PARP1* relevance and overexpression in B-ALL might be a result of a genetic program inherited from the B-cell lineage. This possibility is further reinforced by the higher expression of PARP1 in B cells rather than in myeloid cells and by the downregulation of *PARP1* in CML. 

## 4. Discussion

A major characteristic of *BCR::ABL1* is the induction of genomic instability associated with its leukemogenesis through the promotion of non-conservative DNA repair pathways and consequent increase in tumor mutation burden [8]. As such, through in vitro analyses, we sought evidence that supported the pharmacological redirection of PARP inhibitors, focusing on PARP1 due to its prominent role in DNA repair signaling [10] for the treatment of ALL *BCR::ABL1* p190+ models through the exploration of synthetic lethality pathways.

Both cell lines Raji and Namalwa have a significantly lower expression of PARP1 compared with the ALL cell line, SUP-B15, and the lack of response to the treatment with PARPi is in accordance with the previously reported dependence of PARP1 for proper PARPi cytotoxic effectiveness [54], although Raji still overexpresses in comparison to the non-neoplastic control (Figure 1). It should also be taken into consideration that, while all of the neoplastic cell lines utilized in this study are B-cell neoplasms with lymphoblast-like morphologies and, as such, are expected to share neoplastic characteristics in vitro, both Raji and Namalwa are representative of Burkitt’s lymphoma (BL). Although BL cases usually present a well-defined cytogenetic abnormality, in the translocation of the *MYC* gene, they are also much more complex and heterogenous than ALL tumors, thus being under the influence of diverse oncogenic pathways that may have redundant effects in maintaining cell survival [55].

Further exploring oncogenic pathways which are common to both cell lines and may help to overcome PARPi cytotoxicity, we highlight the presence of mutations in *TP53*, reported by the biobank DepMap Portal (https://depmap.org/portal (accessed on 20 June 2023)), which correspond to the amino acids R213Q and R248Q of the TP53 protein in Raji and Namalwa, respectively. In a study by Zhang et al. [56], R213Q is demonstrated as responsible for downregulating TP53 activity in the transcription of cyclin dependent kinase inhibitor 1A (CDKN1A), which is an important cofactor in the regulation of cell cycle cascade [57], diminishing cell cycle arrest in response to induced DNA damage. In a similar fashion, mutations encompassing residue R248 of TP53 directly affect its DNA-binding domain, significantly reducing the protein affinity for nucleic acids and the frequency in which TP53 associates in tetramers with DNA [58]. As such, while in breast cancer models the presence of wild-type TP53 is a predictor of resistance to the use of PARPis [59], and the presence of mutations in the DNA-binding domain induces greater levels of PARylation and increased sensitivity to treatment with PARPi talazoparib [60], the same findings must not be extrapolated to models of lymphomas.

Another characteristic shared among both cell lines of BL is the infection of Epstein-Barr virus (EBV), highly associated with the leukemogenesis of B-lymphocyte malignancies, being a major prediction factor of bad prognosis in clinical practice [61,62]. Morgan et al. [63] recently demonstrated the activity of PARP1 in regulating the transcription of viral EBV genes, modulating profiles associated with active and latent infection phases, and although previously proposed by the same research group as a potential treatment strategy for EBV-positive tumors [64], the treatment with PARPi did not show any clinically relevant cytotoxic activity against either cell line.

When compared with the other two analyzed cell lines, SUP-B15 does significantly overexpress PARP1 and was extremely responsive to treatment with PARPi in levels comparable to those of the gold standard Imatinib (Table 1 and Table 2). The translocation generating *BCR::ABL1* that drives Ph+ tumors such as SUP-B15 is described as able to induce leukemia onset and development by itself [65], and Ph+ tumors tend to express simple karyotypes, being overly dependent on *BCR::ABL1* expression, a condition denominated “oncogene addiction” [34]. In this situation, and considering the genomic instability associated with *BCR::ABL1* presence [8], the inhibition of PARP through the use of AZD2461, with consequent deregulation of DDR pathways, was shown to be highly disruptive to the maintenance of cell homeostasis, inducing considerable loss of viability in this model representative of Ph+ ALL.

Although AZD2461 does not have any tolerability tests in patients with published results, its use in vitro does not show cytotoxicity over the viability of B-lymphocytes, even in concentrations as high as 40 µM [66]. Therefore, this low cytotoxic profile observed in non-neoplastic cells reinforces the PARPi specific activity in the exploration of synthetic lethality pathways [67] and corroborates with the proposition that phenotypic alterations associated with the leukemogenesis of *BCR::ABL1* p190+ B-cell ALL are determinant factors to the sensitivity to PARPis and not the inherent physiology of B-lymphocytes per se.

Our cell cycle analysis reveals important cytostatic characteristics of both AZD2461 and imatinib, with similar profiles of significant arrest in G0/G1 in detriment of cells in phase S (Figure 2). In the cell death assay, however, it is clear that AZD2461 induces significant early apoptosis when compared both to the untreated as well as to the imatinib treated experiments (Figure 3), but the respective pharmacodynamics should be taken into consideration for proper data interpretation. Although IC50 values of both drugs when treating SUP-B15 are similar, AZD2461 is an inhibitor of enzymes that play a direct role in cell homeostasis maintenance [68], showing high cytotoxic activity shortly after treatment administration, while imatinib acts by inhibiting BCR::ABL1, an important transcriptional factor [69], exerting cytotoxicity only after downregulation of downstream effectors in the signaling cascade.

Furthermore, the higher frequency of death marker presentation in assays treated with AZD2461 may reflect its activity over the previously mentioned tumoral subpopulations detected in the cell cycle analysis. Comparing the results from the cell cycle arrest and early death experiments, it may be interpreted that the cells marked precociously with annexin-V are representative of the same subpopulation that drastically diminishes in frequency in the cell cycle arrest analysis, indicating that AZD2461’s ability to induce early death does not encompass the whole of the SUP-B15 cell line, but rather its subpopulations. From the obtained results, characterization of the subpopulations which form the cell line through single-cell transcriptomic analysis or differential marking of membrane antigens is of extreme importance for proper stratification of risk groups inside this leukemic model and for the prediction of PARPi effectiveness.

The observation of high *PARP1* expression in the SUP-B15 cell line is corroborated by the findings of our aCGH array analysis, revealing amplification of the long arm of chromosome 1, where *PARP1* is localized (Figure 6). Additionally, with the functional annotation analysis it was possible to identify gains involving molecular mechanisms referring, for the most part, to biological processes of the immune system (Figure 7). The PARP pathway plays an important role in the regulation of hematopoietic cell differentiation and in several functions of the immune system, as already widely discussed in the literature [70,71,72,73,74]. Among the main functions are the maintenance of homeostasis related to the differentiation and maturation of B-cells, where the deficiency of the PARP1/2 pathway in these cells demonstrates an increased rate of DNA damage and an accelerated process of apoptosis, pointing to the important role of this pathway in the maintenance of genomic stability in hematopoietic cells and components of the immune system [52]. This critical role in B-cell development elucidates the high expression of *PARP1* seen in this population through single-cell transcriptomic analysis (Figure 9A) and the gains in immune system function of the cell line overexpressing *PARP1*. Furthermore, studies have demonstrated that the therapeutic association of PARPi with immune system pathway inhibitors is indeed promising in several types of tumors [75,76,77,78].

Also, of interest is the observed amplification of the long arm of chromosome 8, encompassing *MYC* proto-oncogene (Figure 6). While *MYC* expression and influence was not investigated in our study, it has been previously reported as a mediator of PARP1 activity, inducing increased activity of nonhomologous end-joining pathways [79,80]. In multiple myeloma models, Caracciolo et al. [81] demonstrated the effectiveness of PARPi olaparib in the treatment of MYC-proficient cells, with a similar pattern of DNA-damage and cell death seen after PARP1 knockout, suggesting a synthetic lethality model correlating the axis MYC-PARP1. However, on in vivo models of *MYC*-driven B-cell lymphoma, Galindo-Campos et al. [82] report different roles for both PARP1 and PARP2 on tumor progression, where PARP1 depletion is associated with increased frequency of T cells, proinflammatory phenotypes and consequent accelerated tumorigenesis, while PARP2 depletion halts B-cell expansion through DNA damage-mediated death. Thus, *MYC* mutational status and expression profile may also be a cofactor underlying the sensitivity of our model of p190+ ALL to PARP inhibition and future investigations are needed to fully elucidate this hypothesis.

Finally, the validation of the differential expression of *PARP1* in p190+ ALL patient samples is attested when looking both into our molecular profile analysis and into online patient sample datasets (Figure 8), corroborating with the translational character of our study and reinforcing the potential of PARP1 as a biomarker in this subset of patients. The low levels of *PARP1* in p190+ CML, however, indicate that the presence of the p190 isoform alone is not enough to dictate high *PARP1* expression, although the use of PARPis has already been demonstrated to have cytotoxic effect over *BCR::ABL1* positive CML in vivo, even though PARP1 levels were not quantified [83]. Altogether, a larger patient cohort is still necessary to establish stronger associations correlating high *PARP1* expression and patient’s clinical features such as age, WBC, blast percentage, immunophenotype, and karyotype.

## 5. Conclusions

In this exploratory study, high PARP1 expression and presence of *BCR::ABL1* p190+ translocation were used as predictors of PARPi sensitivity in cohorts of B-cell neoplastic cell lines, with the observed similar cytotoxic profiles of PARPi, AZD2461, and gold standard treatment, imatinib, against a *BCR::ABL1* p190+ ALL cell model fomenting the idea of utilizing PARP as a potential therapeutic target in this neoplastic model. Still, further in vivo studies and more extensive characterization of other *BCR::ABL1* p190+ models are necessary to validate our hypothesis. Overall, we hope our findings help expand the characterization of molecular profiles in ALL settings and guide future studies into novel biomarker detection and pharmacological choices in clinical practice.

## Figures and Tables

**Figure 1 cancers-15-05510-f001:**
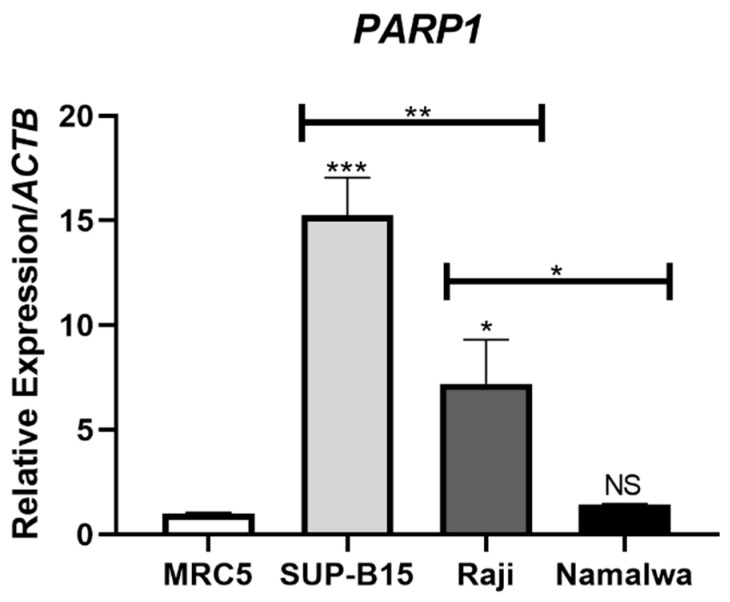
Overexpression of *PARP1* in the SUP-B15 cell line compared with other lymphoblastic cell models. *PARP1* expression was normalized through the endogenous control *ACTB* and the non-neoplastic cell line MRC5 was used as the calibrator. Statistical differences were analyzed through ANOVA followed by Bonferroni’s multiple comparisons. NS: Not significant; * *p* < 0.05; ** *p* < 0.01, *** *p* < 0.0001.

**Figure 2 cancers-15-05510-f002:**
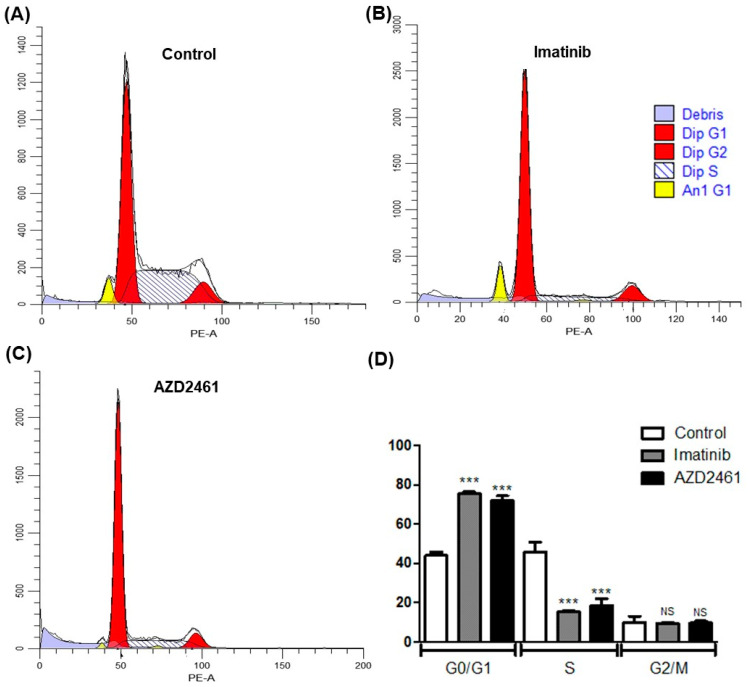
Cell cycle arrest profile in the SUP-B15 cell line after treatment with either AZD2461 or imatinib. Cells were treated with sub-inhibitory concentrations of 1.5 µM of either AZD2461 or imatinib in a 24-h incubation period and the graphs represent the means from three distinct experiments. Cell cycle arrest profile was compared among the treated experiments and the non-treated control through ANOVA followed by Bonferroni’s multiple comparisons. Fluorescence emission was detected after sample processing with DNA intercalating agent, propidium iodide (PI), and is represented in the X-axis, while the Y-axis displays the number of events analyzed. (**A**) Control experiment treated with equivalent volume of DMSO. (**B**) Imatinib treated experiment. (**C**) AZD2461 treated experiment. (**D**) Percentage of cells in each phase of the cell cycle after the treatment and statistical comparison with the non-treated control. Dip: Pseudodiploid population; An1: Hypodiploid population; NS: Not significant; *** *p* < 0.0001.

**Figure 3 cancers-15-05510-f003:**
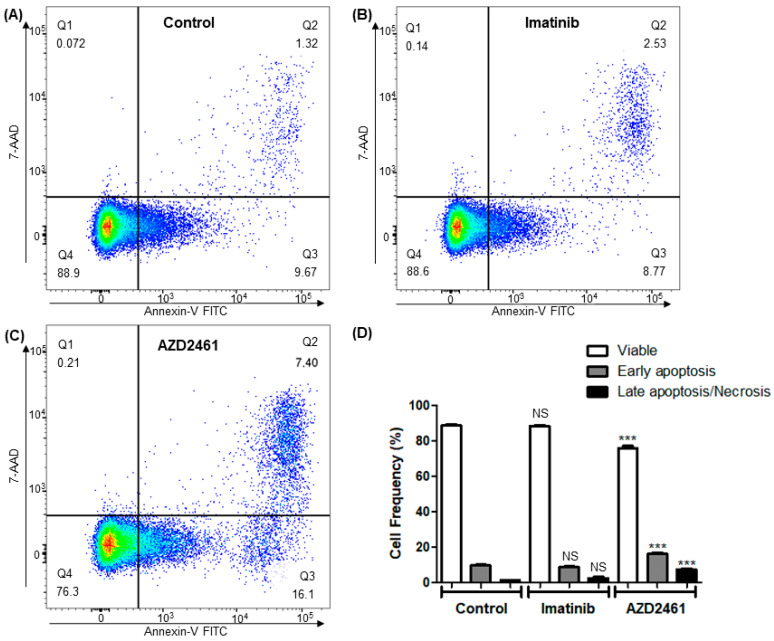
Early apoptosis induction in the SUP-B15 cell line after treatment with AZD2461 or imatinib. Cells were treated with sub-inhibitory concentrations of 1.5 µM of either AZD2461 or imatinib in a 24-h incubation period and the graphs represent the means from three distinct experiments. Events are displayed as pseudocolor plots, with warmer tones representing increased number of events. Cell cycle arrest profile was compared among the treated experiments and the non-treated control through ANOVA followed by Bonferroni’s multiple comparisons. (**A**) Control experiment treated with equivalent volume of DMSO. (**B**) Imatinib treated experiment. (**C**) AZD2461 treated experiment. (**D**) Population frequencies and significance of variations. NS: Not significant; *** *p* < 0.0001.

**Figure 4 cancers-15-05510-f004:**
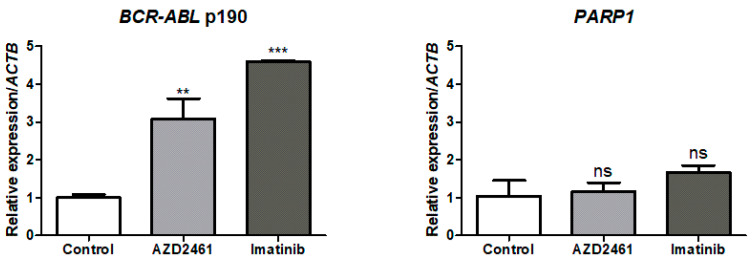
Expression levels of the biomarker *BCR::ABL1* p190 and the *PARP1* transcript in the SUP-B15 cell line after treatment with AZD2461 or imatinib. Cells were treated with sub-inhibitory concentrations of 1.5 µM of either AZD2461 or imatinib in a 24-h incubation period and the graphs represent the means from three distinct experiments. *BCR::ABL1* p190 and *PARP1* expression were normalized through the endogenous control *ACTB*. Expression levels were measured in the SUP-B15 cell line comparing the non-treated control experiment and the experiments after the proposed treatments. Statistical differences were analyzed through ANOVA followed by Bonferroni’s multiple comparisons. ns: Not significant; ** *p* < 0.01, *** *p* < 0.0001.

**Figure 5 cancers-15-05510-f005:**
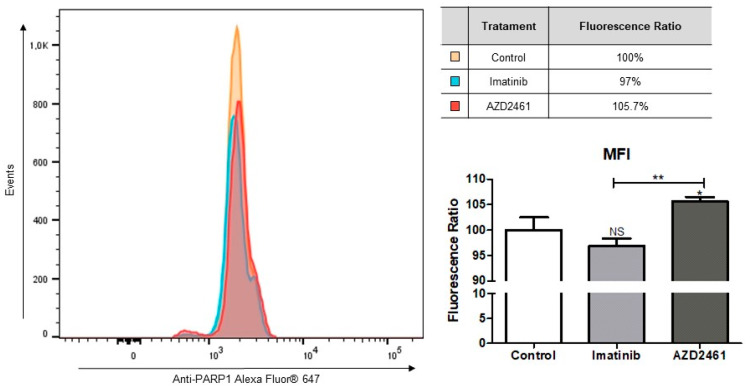
Treatment with AZD2461 reveals cell populations with increased PARP1 protein. Cells were treated with sub-inhibitory concentrations of 1.5 µM of either AZD2461 or imatinib in a 24-h incubation period and the graphs represent the means from three distinct experiments. PARP1 levels were measured through the mean fluorescence intensity (MFI) of the fluorochrome Alexa Fluor^®^ 647 and compared between the treated experiments and the non-treated control through ANOVA followed by Bonferroni’s multiple comparisons. MFI ratios are represented as percentages relative to 100% of fluorescence of the non-treated control. NS: Not significant; * *p* < 0.05; ** *p* < 0.01.

**Figure 6 cancers-15-05510-f006:**
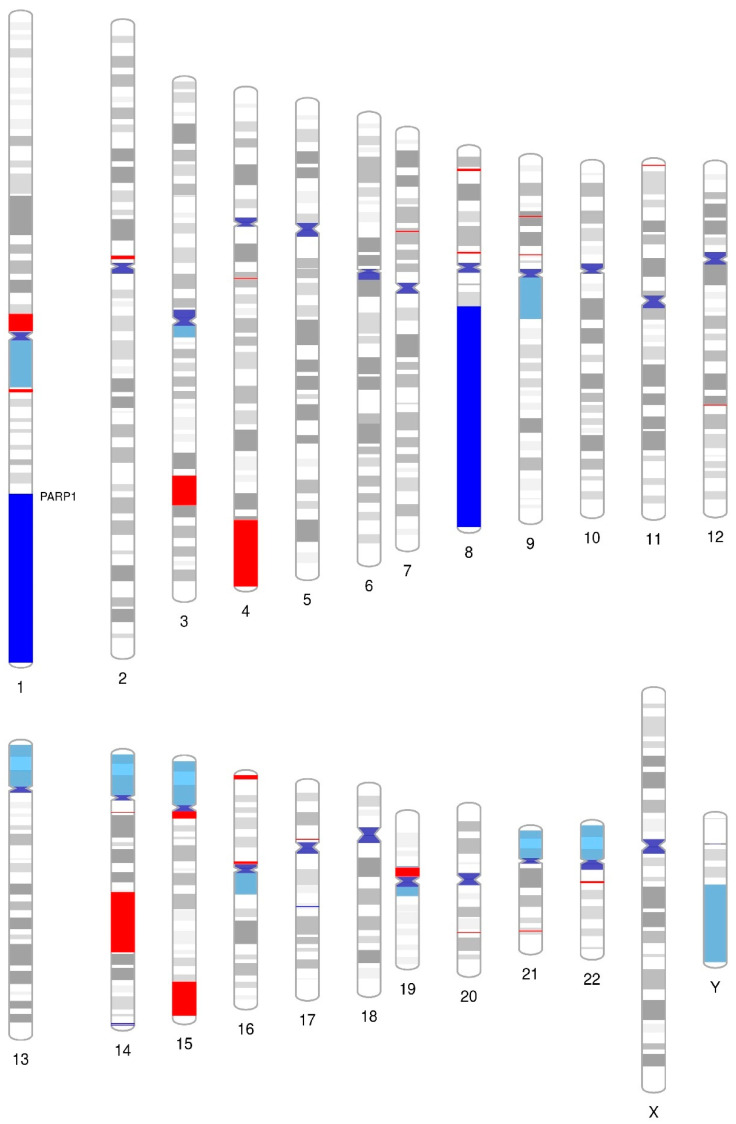
Chromosome’s ideogram showing CNAs identified in the SUP-B15 cell line. Dark blue indicates gains, red represents losses, and light blue represents condensed heterochromatin.

**Figure 7 cancers-15-05510-f007:**
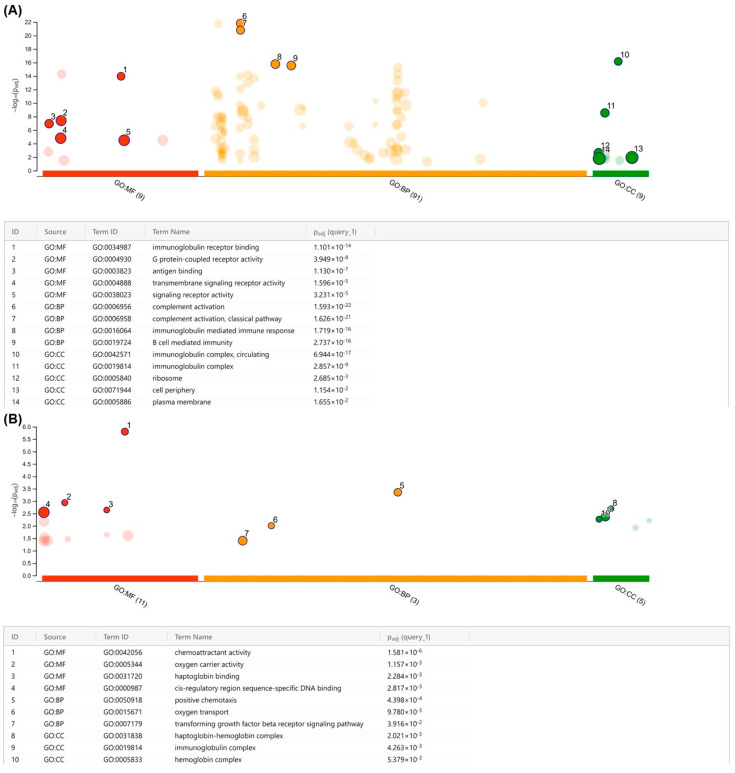
Functional gene ontology (GO) annotation of CNAs from the SUP-B15 cell line. Key significant terms enriched in gene (**A**) gains and (**B**) losses are annotated in the table below. The *x*-axis displays the functional terms and the *y*-axis shows −log10 of the FDR-adjusted *p*-value from the enrichment test. (GO: MF) molecular function, (GO: BP) biological process, (GO: CC) cellular component. GO size represents the number of total genes in each specific ontology.

**Figure 8 cancers-15-05510-f008:**
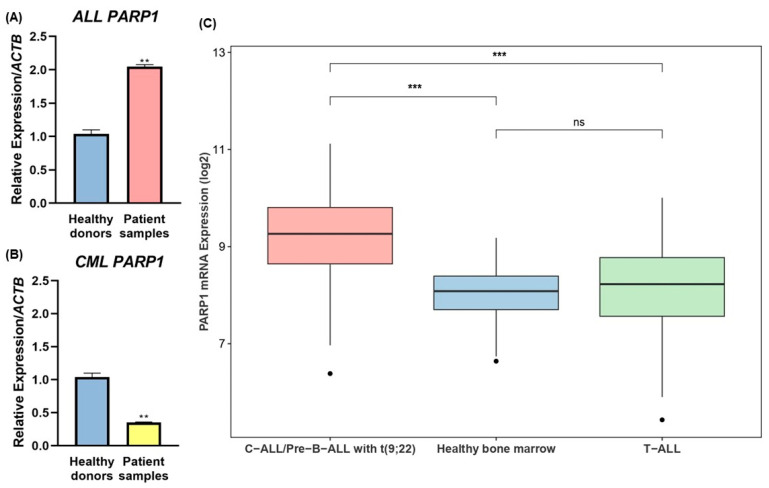
*PARP1* expression in *BCR::ABL1* p190+ patient samples. (**A**) Pooled analysis of the average fold change of *PARP1* expression in ALL p190+ patient samples. (**B**) Pooled analysis of the average fold change of *PARP1* expression in CML p190+ patient samples. (**C**) *PARP1* expression in cohorts of ALL patients from the GSE13159 dataset. Healthy blood donors were used as comparative controls. Statistical differences were analyzed through the Student’s T-test or the Wilcoxon test. ALL: Acute lymphoblastic leukemia; C-ALL: Childhood Acute Lymphoblastic Leukemia; CML: Chronic myeloid leukemia; ns: Not significant; ** *p* < 0.01; *** *p* < 0.0001.

**Figure 9 cancers-15-05510-f009:**
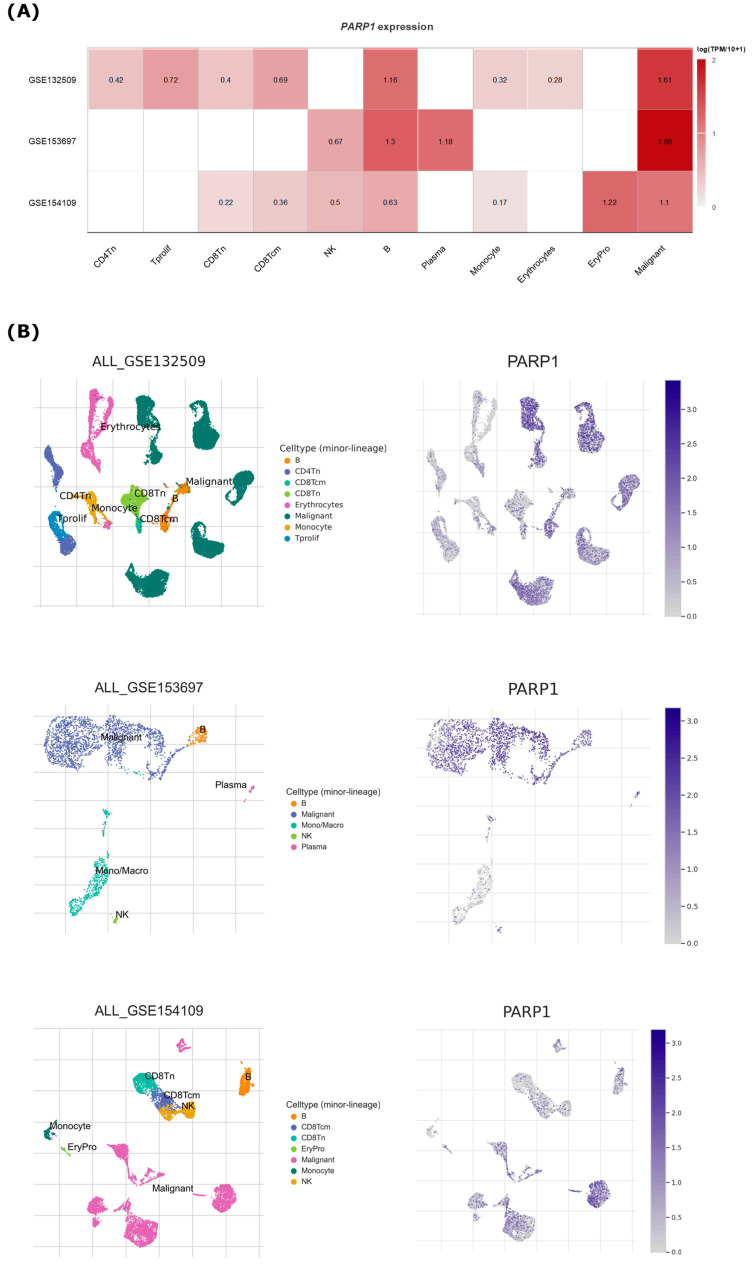
Single-cell analysis of *PARP1* expression in c-ALL datasets. (**A**) Heatmap of *PARP1* expression among cell subtypes in each dataset. (**B**) Single-cell view of cell populations detected and overlapping *PARP1* expression in each dataset.

**Table 1 cancers-15-05510-t001:** Minimum inhibitory concentrations for 50% of the cells (IC50) after 72-h incubation periods. Values were measured in nanomolar (nM) units.

	Cell Line	SUP-B15	Raji	Namalwa
Drug				
AZD2461		344.3 nM (240.1–493.7) (R^2^: 0.9358)	NR	20,938 nM (16,152–27,143) (R^2^: 0.9806)
Imatinib		329.2 nM (215.7–502.2) (R^2^: 0.9385)	2509 nM (1622–3882) (R^2^: 0.9431)	1296 nM (803.8–2089) (R^2^: 0.9284)
Doxorubicin		20.32 nM (11.41–36.19) (R^2^: 0.9233)	85 nM (65.13–110.9) (R^2^: 0.9694)	75.93 nM (49.63–116.2) (R^2^: 0.9622)

NR: Not Reached; R^2^: Coefficient of determination.

**Table 2 cancers-15-05510-t002:** Minimum inhibitory concentrations for 50% of the cells (IC50) of the SUP-B15 cell line after 24 or 48 h of incubation. Values were measured in nanomolar (nM) units.

	Period	24 h	48 h
Drug			
AZD2461		3925 nM (2502–6157) (R^2^: 0.8763)	1421 nM (766.9–2632) (R^2^: 0.8087)
Imatinib		3966 nM (2398–6558) (R^2^: 0.8612)	1230 nM (747.5–2024) (R^2^: 0.8905)
Doxorubicin		141.6 nM (93.92–213.6) (R^2^: 0.9305)	57.33 nM (36.53–89.97) (R^2^: 0.9342)

R^2^: Coefficient of determination.

## Data Availability

The datasets analyzed during this current study are available from the corresponding author on request. Patient’s clinical characteristics and qPCR data are available in the Appendix A Files. The genomic data acquired through the aCGH technique is publicly available in Gene Expression Omnibus (GEO) at GSE239416.

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
