# Peer review of "PARP1 Characterization as a Potential Biomarker for BCR::ABL1 p190+ Acute Lymphoblastic Leukemia"

_cancers, 2023, doi:10.3390/cancers15235510_

Round 1

Reviewer 1 Report

Comments and Suggestions for Authors

In the manuscript “Targeting PARP as a potential therapeutical strategy in BCR::ABL1 p190+ Acute Lymphoblastic Leukemia: Biomarker characterization in preclinical and clinical models” Authors demonstrate the effectiveness of PARPi in the treatment of BCR::ABL1 p190+ ALL cell models and that PARP1 is differentially expressed in patient samples. The manuscript requires a considerable editing before acceptance for publication.

There are following major concerns with the manuscript:

1)     Please shorten the title of manuscript. Currently, it’s too long.

2)     Please provide catalogue numbers for all the drugs used in the study.

3)     Please provide statistical analysis for all the experiments conducted in the study. Currently this section is  missing.

4)     Please avoid using term “PARPi” until defined/specified clearly in manuscript.

5)     Please mention the conclusions from Figure6-9 in context of current manuscript. This feels very general and not much reasoning is provided on why they are in manuscript.

6)     Figure 1and table 1 is unnecessary and lacs novelty. Rather these details can be easily described in method and material section.

7)     Please check the time course dependent changes in expression of PARP1 in figure 4. Also, in the graph they need to mention the name of gene authors are trying to access.

8)     Detailed figure legend is missing in Figure 1-5.

9)      Please consider revisiting manuscript in context of grammar and English.

10)  In heading 2.2, Please use “drug treatment/chemical treatments than saying chemical substances.

11)  Please provide more details on human patient samples. Also, authors need to specify very clearly that what sub-fraction of human blood/samples they are investigating in the study. Currently, it is not very clear.

12)  Also, Discussion section is too long need proper citation of more relevant references for understanding and conclusion of hypothesis proposed in manuscript.

Comments on the Quality of English Language

Please see comment 9

Author Response

Dear Reviewer,

The attachment contains all our answers to your suggestions

King Regards!

Reviewer 2 Report

Comments and Suggestions for Authors

In the manuscript entitled “Targeting PARP as a potential therapeutical strategy in BCR::ABL1 p190+ Acute Lymphoblastic Leukemia: Biomarker characterization in preclinical and clinical models”, Machado and colleagues investigated the effectiveness of PARPi as a therapeutic option of BCR::ABL1 p190+ ALL cell models. They contributed to the broadening of molecular profile understanding within ALL contexts and provided a direction for future inquiries regarding the identification of new biomarkers and pharmaceutical options in clinical practice. This potential treatment for ALL p190+ isoform will hold great significance in the field. However, I have listed below a series of questions and suggestions that might further improve the quality of the manuscript. These concerns must be addressed before the publication of this paper.

1. Some figures are not connected with their figure legends. Please adjust the format.

2. In the Figure 1, what are the PARylation levels in these cell lines, are they correlating with PARP1 expression? Is it possible to examine the PARP activity by WB or immunofluorescence? Similarly, it’s interesting to know the PARP activity after the treatment in Figure 5.

3. Why the authors didn’t show the status of BRCA or homologous recombination? It’s important to identify the DNA repair proficiency or deficiency in ALL p190+ background when related the sensitivity to PARPi.

4. Why there are no supplementary tables?

5. There are no detail interpretation of Figure 9A and 9B. If possible, please further explain the single-cell analysis of PARP1 expression. It would be better to label the cALL and B-ALL near the ID to help the readers better understand the results.

6. Minor questions:

Languages issues/typos: line 75, “poli-ADP” should be “poly-ADP”, etc.

In Figure 2, what did those black triangles under X-axis indicate for? What’s the label for Y-axis of 2A-2D?

In Figure 5, should the “MFL” be replaced by “Increased PARP1 population” or other precise labels?

Please double check all the details in the manuscript.

Author Response

Dear Reviewer,

The attachment contains all our answers to your suggestions.

Kind Regards!

Reviewer 3 Report

Comments and Suggestions for Authors

The manuscript "Targeting PARP as a potential therapeutical strategy in BCR::ABL1 p190+ Acute Lymphoblastic Leukemia: Biomarker characterization in preclinical and clinical models", written by Machado et al. describes the effects of PARP inhibition on three leukemic cell lines, in comparison with the effects of imatinib, standard tyrosine kinase inhibitor used for leukemia treatment. Experiments include analysis of PARP1 expression in cell lines, growth inhibition analyses of PARP inhibitor AZD2461, imatinib and doxorubicin, cell cycle profiles of drug-treated B-ALL cell line, analysis of apoptosis and expression of BCR-ABL and PARP1 after treatment, as well as analysis of chromosomes losses and gains in analyzed cell line. In addition, analysis of PARP expression in BCR::ABL patient samples, from database cohorts, was done.

The Introduction presents the data important for the content. Materials and methods are well described, and results well and clearly presented. Although Discussion tries to explain the data, it seems to me that the explanations are superficial, based on only some of the data, and missing deep causal reasons. Therefore, also Conclusions (and the title), suggesting using PARP as a biomarker and pharmacological choice in the clinical practice should be strengthen with more evidence.

The experiments were done on three cell lines, but only one of them was B-ALL. If the aim was to show that PARP inhibitor can influence BCR-ABL p190 translocation positive B-ALL, as a targeted therapy, several such lines should be analyzed, possibly with different mutations or with and without mutations. There are literature data that some other B-ALL cell lines did not show loss of viability after treatment with (other) PARP inhibitors.

The manuscript lacks more biological view of the processes. Possible cell sensitivity on PARP inhibitors is a consequence of its influence on different signaling pathways, different in each cell line (as PARP modifies many signaling molecules, takes part in DNA damage repair, but also in differentiation). It is known that PARP has a role in blood cell differentiation and its expression can depend on the differentiation status. In that sense different levels of PARP can be explained by different cell lines and cell types. PARP could have different roles in different cell types and processes, and thus cellular microenvironment could determine the sensitivity on the PARP inhibitors, not only the total expression of PARP. Many cell types retain cell viability after treatment with PARP inhibitors, but these relations cannot be compared with that of TKI resistance and BCR-ABL expression, as the mechanisms are different. Also, comparisons in Discussion considering expression level of PARP (629-632) need more evidence. Levels of PARP on expression level in fact should also be commented in the light of its function: PARP exerts its function in DNA damage repair when activated by DNA damage, and PARP inhibitors are not supposed to inhibit its expression, but activation. And PARP can have certain functions also through protein-protein interactions, even not being activated.

Other comments:

sentence correction: lines 123, 518, 540, 589,

Figure legend should be attached to the Figure, not written as a part of the main text.

Comments on the Quality of English Language

sentence correction: lines 123, 518, 540, 589,

Author Response

(The authors gave the same response as above.)

Round 2

Reviewer 1 Report

Comments and Suggestions for Authors

In the updated manuscript “Targeting PARP as a potential therapeutical strategy in BCR::ABL1 p190+ Acute Lymphoblastic Leukemia: Biomarker characterization in preclinical and clinical models”,the authors did attempt to address all the previous concerns and now the manuscript is convincing and would help advance the understanding of basic of PARP as a potential therapeutical strategy in Acute Lymphoblastic Leukemia and recognition specific characteristics markers in preclinical and clinical models. The manuscript is now well-updated, and I recommend this article for publication.

Author Response

Dear Reviewer,

Kind Regards!

Reviewer 2 Report

Comments and Suggestions for Authors

The authors have largely responded and/or addressed the major concerns. The manuscript quality was significantly elevated. 

However, I listed below some suggestions for further revising the manuscript. When solved, I have no hesitation to suggest the publication of this paper.

Please still double check the details and new revisions including but not limit to:

Line 82, is that word “efflux bombs” or “efflux pumps”?

The languages in the discussion part could still be improved for clarity. For example, line 554, “…models of through the exploration of…” is not clear. The sentence from 560 to 566 is too long to understand. Please reform.

Besides, as the discussion parts expand even longer than before. Maybe the authors could consider citing the figures when mentioning the results respectively.

Author Response

Dear Reviewer,

Kind Regards!

Reviewer 3 Report

Comments and Suggestions for Authors

The authors of the manuscript "PARP1 characterization as a potential biomarker for BCR::ABL1 2 p190+ Acute Lymphoblastic Leukemia" responded to my comments, reorganized the Discussion and Conclusions and improved the manuscript.

Comments on the Quality of English Language

lines 83, 559 small mistakes

lines 234, 232... buffy coat

units should be written separately from the numbers

Author Response

Dear Reviewer,

Kind Regards!
